Spiculous skeleton formation in the freshwater sponge Ephydatia fluviatilis under hypergravity conditions

Bart Martijn C. 1 m.c.bart@uva.nl
de Vet Sebastiaan J. 2 3
de Bakker Didier M. 4
Alexander Brittany E. 1
van Oevelen Dick 5
http://orcid.org/0000-0002-8895-0427 van Loon E. Emiel 6
http://orcid.org/0000-0001-9051-6016 van Loon Jack J.W.A. 7
http://orcid.org/0000-0002-3411-3084 de Goeij Jasper M. 1
1 Department of Freshwater and Marine Ecology, Institute for Biodiversity and Ecosystem Dynamics, University of Amsterdam , Amsterdam , The Netherlands
2 Earth Surface Science, Institute for Biodiversity and Ecosystem Dynamics, University of Amsterdam , Amsterdam , The Netherlands
3 Taxonomy & Systematics, Naturalis Biodiversity Center , Leiden , The Netherlands
4 Microbiology & Biogeochemistry, NIOZ Royal Netherlands Institute for Sea Research & Utrecht University , Utrecht , The Netherlands
5 Department of Estuarine and Delta Systems, NIOZ Royal Netherlands Institute for Sea Research & Utrecht University , Utrecht , The Netherlands
6 Department of Computational Geo-Ecology, Institute for Biodiversity and Ecosystem Dynamics, University of Amsterdam , Amsterdam , The Netherlands
7 Dutch Experiment Support Center, Department of Oral and Maxillofacial Surgery/Oral Pathology, VU University Medical Center & Academic Centre for Dentistry Amsterdam (ACTA) & European Space Agency Technology Center (ESA-ESTEC), TEC-MMG LIS Lab, Noordwijk , Amsterdam , The Netherlands
Pawlik Joseph
Electronic publication date: 2019 Jan 4
Publication date: 2019
Volume: 6
Electronic Location ID: e6055
Received 2018 Aug 30; Accepted 2018 Oct 30
Copyright: © 2019 Bart et al.
Copyright year: 2019
Copyright holder: Bart et al.
License: This is an open access article distributed under the terms of the Creative Commons Attribution License, which permits unrestricted use, distribution, reproduction and adaptation in any medium and for any purpose provided that it is properly attributed. For attribution, the original author(s), title, publication source (PeerJ) and either DOI or URL of the article must be cited.
License URL: https://creativecommons.org/licenses/by/4.0/

Keywords: Freshwater sponges, Gemmule, Spicules, Hypergravity, Skeleton construction

Funding: European Space Agency Spin Your Thesis! educational program ESA-DGC-DET-2014-1218 This project was financed by the European Space Agency Spin Your Thesis! educational program (ESA-DGC-DET-2014-1218). The funders had no role in study design, data collection and analysis, decision to publish, or preparation of the manuscript.

==============================
Successful dispersal of freshwater sponges depends on the formation of dormant sponge bodies (gemmules) under adverse conditions. Gemmule formation allows the sponge to overcome critical environmental conditions, for example, desiccation or freezing, and to re-establish as a fully developed sponge when conditions are more favorable. A key process in sponge development from hatched gemmules is the construction of the silica skeleton. Silica spicules form the structural support for the three-dimensional filtration system the sponge uses to filter food particles from ambient water. We studied the effect of different hypergravity forces (1, 2.5, 5, 10, and 20 × g for 48 h)—as measure for environmental stress—on the ability of developing sponges to set-up their spiculous skeleton. Additionally, we assessed whether the addition of nutrients (i.e., dissolved 13C- and 15N-labeled amino acids) compensates for this stress. Our results show that freshwater sponges can withstand prolonged periods of hypergravity exposure and successfully set-up their skeleton, even after 48 h under 20 × g. Developing sponges were found to take up and assimilate dissolved food before forming a functional filtering system. However, fed and non-fed sponges showed no differences in skeleton formation and relative surface area growth, suggesting that the gemmules’ intrinsic energy fulfills the processes of skeleton construction. Additionally, non-fed sponges formed oscula significantly more often than fed sponges, especially under higher g-forces. This suggests that the eventual formation of a filtration system might be stimulated by food deprivation and environmentally stressful conditions. These findings indicate that the process of spiculous skeleton formation is energy-efficient and highly resilient. The uptake of dissolved food substances by freshwater sponges may contribute to the cycling of dissolved organic matter in freshwater ecosystems where sponges are abundant.

Introduction

Sponges are among the oldest—more than 635 million years old—still existing metazoans on Earth (Müller, 1998; Love et al., 2009). Due to its ancient heritage, the sponge body plan is traditionally seen as the original blueprint for multicellularity (Müller et al., 2004; Nosenko et al., 2013). Although sponges lack organs, they possess a high level of functional complexity (Leys, Nichols & Adams, 2009; Srivastava et al., 2010). They have a well-developed food uptake and waste disposal system constructed of numerous small inflow openings (ostia) and chambers containing filter cells (choanocytes) with actively beating flagella to create an internal water flow. Sponges can contract their ostia and water channels to modulate this water flow (Elliott & Leys, 2007; Ludeman et al., 2014). After the indrawn water has passed the choanocyte chambers, waste products are discarded through excurrent canals (oscula). Sensory cilia inside the osculum use calcium channels to adapt the sponge’ water filtering capacity, for example, in response to temperature changes or increased suspended sediment (Ludeman et al., 2014; Cavalier-Smith, 2017).

The freshwater sponge Ephydatia fluviatilis (Porifera, Demospongia, Spongillidae), the species used in this study, is cosmopolitan and found throughout Earth’s entire northern hemisphere (Van Soest et al., 2017). This shows the flexibility of this sponge to endure a wide range of environmental conditions. The colonization capacity of inland waters by freshwater sponges largely depends on the formation of small (∼300 μm diameter) dormant sponge bodies (gemmules) under adverse conditions (Manconi & Pronzato, 2008; Funayama, 2013). This form of asexual reproduction allows the sponges to overcome critical environmental conditions (e.g., low water temperatures during wintertime or desiccation during hot summers) (Manconi & Pronzato, 2016). When conditions are more favorable, a fully-developed miniature sponge can be re-established from a gemmule in approximately 1 week, depending on environmental conditions (Höhr, 1977; Ilan, Dembo & Gasith, 1996; Funayama et al., 2005) (Fig. 1A). First, during gemmule germination, totipotent stem cells contained within the gemmule differentiate into different cell types (Wierzejski, 1915, 1935; Höhr, 1977). Secondly, during a process called hatching, the cells within the gemmule migrate outwards while continuing to differentiate into various cell types–including basal epithelial cells that attach the sponge to its substrate (Rozenfeld, 1970; Höhr, 1977; Harrison et al., 1981). Eventually, the migrated cells proliferate and differentiate into all types of cells to form a fully functional sponge (Höhr, 1977; Funayama et al., 2005).

Figure 1 Developmental stages of germination in the freshwater sponge Ephydatia fluviatilis and hypothetical working model of the effect of hypergravity and nutrient uptake on skeleton construction of a juvenile sponge.

(A) Modified from Funayama et al. (2005). Stage 0: resting gemmule. Stage I: (approximately 2-days post-hatching): cells migrate outward from the gemmule and differentiate into epithelial cells. Stage II: (3–4 days): cells start to proliferate and begin to differentiate into different cell types. This stage includes the development of the spicule-making sclerocytes. Spicule production generally starts around stage II and continues thereafter (Funayama et al., 2005; Mohri et al., 2008). Stage III: (4–5 days post-hatching): the canal system and choanocyte chambers are starting to form. Stage IV: (6–7 days post-hatching): The osculum is created approximately 1 week after hatching, creating a fully functional sponge. (B) Hypothesized response of the influence of hypergravity on developing sponges in combination with feeding. We hypothesize sponges will not be able to set up their spicule skeleton after prolonged exposure to increasing levels of hypergravity, but counteract this inability after additional feeding on dissolved food sources before forming a fully functional active filter-feeding system.

A key process in freshwater sponge development from hatched gemmules is the formation and construction of the skeleton. The sponge skeleton forms the structural support for the three-dimensional filtering system of the sponge (reviewed by Uriz et al., 2003). In demosponges, the skeleton consists of individualized elements (silica spicules), embedded in a fibrous organic matrix made from chitin and spongin (Larsen & Riisgård, 1994; Uriz et al., 2003; Ehrlich et al., 2013). After the spicules are set in an upright position, the filter system of the sponge develops and the sponge can start to obtain nutrients by active filter-feeding (Funayama et al., 2005).

Recently, it was shown for E. fluviatilis that the silica spicules are not randomly distributed throughout the developing sponge body, but are deliberately set-up in a highly-organized manner (Nakayama et al., 2010, 2015), similar to pitching a tent. Spicules are produced by cells termed sclerocytes, at various locations within the hatching gemmule, and transported by so-called transport cells to their final position, where they are erected and cemented by a third type of specialized cells (the basopinacocytes) (Nakayama et al., 2015). This division of labor between various cell types within the sponge has revealed a fundamentally new mechanism of constructing the three-dimensional body shape of animals (Nakayama et al., 2015). As stated by Nakayama et al. (2015), the spiculous skeleton construction process of E. fluviatilis is the only known biological mechanism in which a sequence of cooperative behaviors of individual cells leads to the active construction of a self-organized biological structure using non-cellular materials. Moreover, the mediation of the different steps of spiculous skeleton construction by specialized cells allows for high plasticity and helps to generate the morphological diversity of fully-grown E. fluviatilis specimens.

Gemmule formation and hatching are seasonally dependent and it appears that external factors such as water temperature, water turbidity, and illumination are responsible for these processes (Harsha, Francis & Poirrier, 1983; Ilan, Dembo & Gasith, 1996). However, at present it is unknown how most of these external factors affect the formation and hatching of gemmules. In addition, gemmule hatching and spicule body construction up to the moment of a fully-developed filter system is presumed to be mediated by internally-stored energy and therefore independent of externally available food sources. Archeocytes within the gemmules of fresh-water sponges are known to contain reserve substances including RNA, lipids, and polysaccharides (Ruthmann, 1965). Hence, germination studies are usually performed in medium without added food sources (Funayama et al., 2005; Elliott & Leys, 2007). This hypothesis is, however, never tested and therefore it is unknown whether developing sponges can obtain external energy sources before they have developed the capacity to actively filter-feed, and if this influences their development.

Access to the large diameter centrifuge (LDC) of the European Space Agency (ESA) in Noordwijk, the Netherlands, enabled us to test the influence of increased environmental stress (i.e., increased hypergravity (g) forces) on the construction of the siliceous skeleton of developing sponges. Hypergravity is an artificially created condition in which the acceleration exceeds the common terrestrial gravitational acceleration of 9.81 m s−2 (1 × g) and this creates a force working directly against the sponge cells erecting the spicules. Furthermore, hypergravity has been shown to alter the intracellular transport and delivery of cell wall material in plants (Chebli, Van Loon & Geitmann, 2012), polyp growth in stony corals (Meroz et al., 2002) and effects skeletal architecture and bone-repair in mammals (Prodanov et al., 2013; Canciani et al., 2015). Exposure to hypergravity acts on the whole cell mass, and cells exposed to several g’s can adapt by decreasing the height of their microtubule network, but increasing the thickness of their actin fibers without affecting cell viability (Kacena et al., 2004; Searby, Steele & Globus, 2005; Van Loon et al., 2009).

We hypothesize that (1) increased g-forces decrease the ability of the sponge cells to transport their spicules to their final location and to erect them due to the higher energy costs involved, preventing the formation of a fully-developed filter system (e.g., no osculum formation) and (2) that food addition will partially compensate for the expected energy shortage caused by hypergravity (Fig. 1B).

To study the effects of external stress on skeleton formation in developing freshwater sponges, we tested how prolonged exposure (48 h) to different hypergravity forces (1, 2.5, 5, 10, and 20 × g) influenced (1) relative surface area increase; that is, the substrate area covered by basal epithelium of the sponge compared to the area that was covered before the experiments commenced as measure of growth, (2) the presence of a set-up skeleton, and (3) the presence or absence of an osculum in developing E. fluviatilis specimens hatched from gemmules. In addition, we tested the ability of the developing sponges under above-mentioned conditions of hypergravity exposure to take up dissolved food (i.e., 13C- and 15N-enriched amino acids) and whether additional feeding affects skeleton set-up and osculum formation.

Materials and Methods

Gemmule harvesting and preparation

Gemmules were collected in the winter months (November 2013–March 2014), from specimens of E. fluviatilis kept in outdoor aquaria at the University of Amsterdam. Sponge tissue containing the gemmules was collected and cleaned by rubbing the sponge tissue between two pieces of corduroy to free the gemmules from the sponge tissue scaffold that can inhibit germination. Detached gemmules were sterilized in 1% H2O2 for 5 min on a shaker at 4 °C to remove bacterial and fungal contaminants included in the coat (Funayama et al., 2005).

Gemmule hatching

A total of 420 gemmules from one individual sponge were plated on sterile 12-well culture plates in sterile M-Medium (Rasmont, 1961; one mM CaCl2, 0.5 mM MgSO4, 0.5 mM NaHCO3, 0.05 mM KCl, 0.25 mM Na2SiO3) under ambient conditions. The plates were sealed by parafilm. Each well of the 12-well culture plates contained one gemmule in approximately four mL of M-medium. After 4 days, before placing the gemmules in the LDC, the individuals that did not hatch were discarded and of the remaining hatched specimens, only the sponges in stage II were selected (i.e., specimens without any erected spicules or osculum, see Fig. 1A for a description of developmental stages). A total of 72% of the plated gemmules hatched (302 out of 420), of which n = 277 sponges developed to stage II that were subsequently used for the experiment in the LDC (Table 1). Before placing the selected gemmules in the LDC, the medium was refreshed to prevent accumulation of waste products in the well plates.

Table 1 Development of E. fluviatilis gemmules under hypergravity exposure (1 × g = 9.81 m s−2).

g-force	1	2.5	5	10	20	Total	Non-fed	Fed	
Fed	No	Yes	No	Yes	No	Yes	No	Yes	No	Yes				
n	24	22	21	20	21	25	19	29	20	26	227	122	105	
Stage III reached (%)	100	95	100	90	95	92	100	93	100	96	96% (±1%)	99% (±1%)	93% (±1%)	
Stage IV reached (%)	79	73	95	60	29	48	79	59	95	46	66% (±7%)	75% (±13%)	57% (±5%)	
Note:

Stage III sponges have erected silica spicules. Stage IV sponges have erected silica spicules and developed an osculum.

Sponge feeding with 13C- and 15N-enriched amino acids

For each gravitational condition (see below), one group (i.e., approximately half) of the gemmules was fed with isotopically-enriched (13C and 15N) dissolved amino acids in order to study nutrient uptake, whereas the other group remained unfed (Table 1). The fed groups (n = 22, 20, 25, 29, 26, respectively, for 1, 2.5, 5, 10, and 20 × g) were randomly selected and received M-Medium with 390 μg L−1 tracer 13C- (11 μmol L−1) and 15N- (2.2 μmol L−1) enriched dissolved amino-acids added prior to the LDC runs. The non-fed group (n = 24, 21, 21, 19, 20, respectively, for 1, 2.5, 5, 10, and 20 × g) did was kept in sterile M-Medium throughout the experiment.

Hypergravity experiment in large diameter centrifuge

Hypergravity experiments were performed using the LDC of the ESA (Noordwijk, The Netherlands) following a predetermined protocol (Fig. 2A). The LDC has a diameter of eight m and comprehends four large arms fitted with outward swinging gondolas. The rotational movement of the arms creates an artificial acceleration field at the well plates positioned inside the gondolas, simulating different g-forces depending on gondola placement on the arms and rotational speed (Fig. 2B). Experiments where performed in two 48-h runs. In the first 48-h run developing sponges were exposed to 1 (i.e., as hypergravity control; placed at the center of the rotating LDC), 2.5 and 5 × g and in the second run to 10 and 20 × g (Fig. 2B). Per hypergravity level, both fed and non-fed sponges were tested simultaneously. To assess the effect of rotation on the development of the gemmules, a control experiment without feeding was performed where gemmules were hatched on a non-rotating lab bench, in addition to the rotating control at 1 × g. As conditions did not vary in the LDC room between the first (2.5 and 5 × g) and the second run (10 and 20 × g) no additional hypergravity control at 1 × g was performed during the second run.

Figure 2 Schematic set-up of the experimental procedure using the large diameter centrifuge (LDC) at the European space agency (ESA), Noordwijk, the Netherlands.

(A) Timeline of the experiment. (B) Configuration of the LDC gondolas (top-view). The LDC has four arms each of which can accommodate two gondolas carrying the well plates with the gemmules. When the centrifuge is spun, the gondolas swing out at an angle (θ) and a hypergravity field (geff) inside the gondolas is created by the centripetal forces due to the rotation.

Sponge surface area measurements

Before and after placement in the LDC, the surface area of the substrate covered by the sponges (in mm2) was determined by light microscopy. We are aware that sponge volume is a better metric for growth of sponges building a three-dimensional structure (i.e., the skeleton). However, we were unable to accurately measure volume and therefore used the increase of substrate area covered by the basal epithelium of the sponge relative to the substrate area covered by the basal epithelium of the sponge before placement in the LDC, as a proxy for sponge size/growth. All sponges were photographed and visually checked to assess substrate area cover, the set-up/no set-up of the sponge skeleton, and the presence/absence of an osculum under a stereoscopic microscope (Olympus SZH-ILLD with infinity1 microscopy camera). Surface area measurements were performed using Image J software (https://imagej.nih.gov/ij/).

Stable isotope analysis of sponges

After LDC exposure and microscopy imaging, the labeled M-medium was replaced with non-labeled M-medium for 3 × 5 min to remove as much residual label as possible. After the last replacement the sponges were left for 30 min in the non-labeled medium to remove any residual label from the sponge surface. Subsequently, sponges were then taken out of the well-plates, rinsed with M-medium and pooled per treatment, freeze-dried, homogenized, and stored at −20 °C in silver boats. Pooling was necessary to ensure that sufficient carbon and nitrogen was available for the stable isotope analysis. Samples were put in silver boats, acidified with 5% HCl to ensure removal of inorganic carbon, oven-dried at 60 °C, pinched closed and stored frozen before analysis on an Elemental Analyser (EA, Thermo Scientific, Waltham, MA, United States, Flash EA 1112 Analyzer) that was coupled to a Delta V isotope ratio mass spectrometer (IRMS) for simultaneous measurement of 13C:12C and 15N:14N ratios. Reproducibility for the EA-IRMS analysis was 0.25‰ for 15N and 0.2‰ for 13C.

The uptake of dissolved amino acid C or N was expressed as uptake of μmol tracer C or N per mmol sponge C or N per d. Rates are calculated from the delta notations obtained from the IRMS as δX (‰) = (Rsample/Rref − 1) × 1,000, in which X is the element (C or N), Rsample is the heavy:light isotope ratio in the sample and Rref is the heavy:light isotope ratio in the reference material (Rref = 0.0111797 for C and Rref = 0.0036765 for N). The atomic fraction of the heavy isotope (F) in a sample is calculated as F = Rsample/(Rsample + 1). The excess (above background) atomic fraction is the difference between the F in an experimental sample and the atomic fraction in a control (i.e., non-enriched) sample: E = Fsample − Fcontrol. The excess incorporation of 13C and 15N was multiplied by 1,000 to express rates in μmol tracer C or N per mmol sponge C or N and divided by the incubation time to convert to daily rates.

Statistical analysis

To analyze the effect of rotation without hypergravity exposure on the sponges, a two sample t-test was performed to compare the means of the two (rotating/non-rotating) 1 × g groups. A Chi-square test was used to compare osculum development between the two groups. Both tests were performed in SPSS version 25 (Released 2011, IBM SPSS Statistics for Windows, Version 25.0; IBM Corp., Armonk, NY, USA).

Surface area cover increase is expressed as a percentage relative to the original surface are covered by each developing sponge hatched from a gemmule in stage II at the start of the LDC experiment. The effect of hypergravity on percentage surface increase was investigated by a linear model, adopting a 0.05 significance level. Besides evaluation of the overall model significance, also the pairwise differences among the treatment-levels were made through a Tukey HSD test with a 95% family-wise confidence level.

Osculum formation was analyzed with a generalized linear model using a binomial error function. To address the effect of osculum formation and g-force on the mean dissolved organic matter (DOM) uptake (measured by 13C and 15N enrichment), an additive linear model was evaluated, using a 0.05 significance level.

In the results, means and proportions are reported in combination with the standard deviation between brackets. All analyses were carried out in R statistics programming environment R 3.3.2 (The R Foundation for Statistical Computing, 2004–2013).

Results

Sponge development under hypergravity exposure

The non-rotating and rotating controls at 1 × g did not show a significant difference in sponge growth (two-sample t-test, t(31) = −0.341, p > 0.1), skeleton formation (in both groups 100% of the sponge specimens showed set up skeleton) or osculum development (Chi-square test, χ2 (2, n = 33) = 1.8397, p > 0.1).

Over all levels of hypergravity exposure, 96% (±1%) of stage II juvenile sponges developed to stage III juvenile sponges (i.e., with erected skeleton), 66% (±7%) of all sponges reached stage IV (i.e., build an osculum), and formed a fully functional sponge during their 48-h treatment (Table 1). All sponges showed an increase in surface area of on average 196% (±70%) compared with the surface area before placement in the LDC. However, the increase in surface area significantly decreased at higher hypergravity levels (R2 = 0.07, n = 227, p < 0.01) (Figs. 3A and 4). Comparing the different treatment levels (using a Tukey HSD test), only the relative surface area change between 1 vs. 20 × g (−47%, p < 0.01) and 5 vs. 20 × g (−45%, p < 0.01) were significant.

Figure 3 Spicule skeleton in developing E. fluviatilis gemmules under different hypergravity forces.

(A) Sponges photographed after the LDC centrifugation. Note that although surface area decreases with gravity level, under all hypergravity forces the sponges managed to produce spicules, set-up their three-dimensional skeleton and form an osculum. (B) Zoomed-in photographs of osculum formation and spicules after 48 h exposure to 20 × g.

Figure 4 Substrate area cover increase in fed and non-fed E. fluviatilis gemmules under different hypergravity conditions.

The average surface area cover increase (% of initial size) of fed (n = 22, 20, 25, 29, 26, respectively, for 1, 2.5, 5, 10, and 20 × g) and non-fed (n = 24, 21, 21, 19, 20, respectively, for 1, 2.5, 5, 10, and 20 × g) sponges under the different hypergravity forces. Error bars represent standard errors of the mean.

No effect of feeding on surface area increase was found (Fig. 4). A significant interaction effect of g-force and feeding on mean osculum formation per group was observed, where fed sponges formed oscula less frequently than non-fed sponges, especially at higher g-force (logit (osculum formation) = 0.71−0.13 feeding + 0.06 g-force − 0.10 feeding g-force; Nagelkerke-R2 = 0.08, n = 227, p < 0.05).

Tracer isotope incorporation

Amino acid assimilation was confirmed by significant enrichment in both δ13C (42.5 ± 11.1‰) and δ15N (1,202.8 ± 280.2‰) in the tissues of all fed sponges compared with background, non-fed, sponge tissue (−30.5‰ δ13C and 18.2‰ δ15N). On average, hatching E. fluviatilis assimilated 0.4 ± 0.1 μmol Camino acids mmol C−1sponge d−1 and 4.3 ± 1.0 μmol Namino acids mmol N−1sponge d−1 (Fig. 5). In total, 9.1% ± 3.7% and 41.2% ± 18.5% of the added amino acid carbon and nitrogen, respectively, was processed by the sponges during the 48-h incubations. No significant effect of g-force was found on 13C-assimilation rates, while the overall effect for 15N assimilation rates was significant (C∼g-force, n = 10, p = 0.165; N∼g-force, n = 10, p < 0.05) (Fig. 5). When comparing the effects on 15N for different treatment levels (using a Tukey HSD test), only the difference between 5 × g vs. 20 × g (−2.7, p < 0.01) appeared significant.

Figure 5 Incorporation of isotopically-enriched (13C and 15N) amino acids in developing E. fluviatilis.

Average incorporation rates are expressed as μmol C or Ntracer mmol−1sponge d−1. n = 22, 20, 25, 29, 26, respectively, for 1, 2.5, 5, 10, and 20 × g. The dotted line represents average uptake across all g force levels. Error bars represent standard errors of the mean. Asterisk indicates significant difference compared to uptake at 1 × g.

Discussion

Spiculous skeleton construction and osculum formation under hypergravity

Sponges are known for their enormous plasticity and opportunistic nature when it comes to food selection (Ribes, Coma & Gili, 1999; McMurray et al., 2016) and regeneration after environmental stress (Ayling, 1983; Wulff, 2010; Alexander et al., 2015). This is particularly important in ecosystems with large seasonal fluctuations in water temperature and food availability, where the formation of gemmules by freshwater sponges is a very successful strategy to survive these adverse and possibly lethal conditions. When conditions are more favorable, gemmules hatch and develop into fully grown sponges in order to enter a successful sexual reproductive stage and obtain sufficient biomass to gemmulate (Funayama et al., 2005; Manconi & Pronzato, 2016). The formation of the spiculous skeleton is a crucial feature of the morphogenesis of freshwater sponges, resulting eventually in the formation of an active filter-feeding system (Rozenfeld, 1980; Funayama et al., 2005), enabling the sponge to grow and reproduce. Albeit the hypergravity forces used in our experiment are not naturally occurring, this is the first in vivo study of freshwater sponges to show that skeleton formation of E. fluviatilis is highly resilient under high levels of stress. Even after the juvenile sponges were exposed to hypergravity forces up to 20 × g for 48 h, they managed to survive, grow, create and organize their spicules, and develop a functional food-uptake system. In comparison, a well-trained human astronaut has a g-force tolerance of approximately 9 × g for only short periods of seconds to minutes (Wu et al., 2012). Our initial hypothesis that increased g-forces decrease the ability of the sponge cells to transport and erect their spicules and prevent the formation of a fully-developed filter system (e.g., no osculum formation) is clearly rejected. Sponges showed a significantly smaller increase of substrate area covered at higher g-force, which may suggest impaired growth, but almost all specimens (96 ± 1%) were able to set-up their spiculous skeleton and the majority was able to create and osculum (66 ± 7%).

It is noteworthy that, although this study focused on the siliceous spicules of the E. fluviatilis skeleton, the fibrous matrix of spongin and possibly chitin in which the spicules are embedded, may also play a role in withstanding environmental stress (Ehrlich et al., 2010, 2013). The presence of chitin is known to be responsible for the rigidification or skeletal structures in invertebrates including marine demosponges (Brunner et al., 2009; Ehrlich et al., 2010) and has since long been known to be an important component of E. fluviatilis gemmules (Zykoff, 1892). However, the presence of chitin has not been reported in adult specimens of E. fluviatilis and it is at present unknown whether or how the fibrous matrix of the skeleton changes in response to environmental conditions.

Fed vs. non-fed sponges under hypergravity

We also show that hatching E. fluviatilis can take up dissolved nutrients (i.e., amino acids). Previous evidence corroborates that freshwater sponges can feed on dissolved organic food sources, such as dissolved proteins (Weissenfels, 1976; Manconi & Pronzato, 2008). However, this study shows, for the first time, that developing sponges hatched from gemmules take up and incorporate dissolved food sources in their tissue possibly even before arranging their filter system and start active pumping. In that case, the dissolved 13C- and 15N-labeled amino acids can be acquired either via passive diffusion or through phagocytosis by choanocytes or surface pinacocytes (Willenz & Van De Vyver, 1982; Willenz, 1980). However, the uptake of dissolved amino acids did not affect skeleton formation nor surface area growth. We therefore conclude that the added nutrients are not a necessity for the developing sponges to overcome the environmental stress caused by hypergravity. Since most of the sponges developed normally even under extreme levels of stress, this suggests that the process of erecting the spiculous skeleton relies mainly on intrinsic energy contained within the gemmule. Interestingly, fed sponges developed oscula significantly less frequently compared with non-fed sponges, especially at exposure to the highest (10 and 20 ×) g-forces. We speculate that under extremely stressful conditions, when energy costs are highest, sponges invest in the formation of an active filtration system to acquire new energy. Additional food can help prolong the process of hatching and skeleton development, which might increase their chances of survival throughout severe stress conditions.

The role of dissolved food in the diet of freshwater sponges

The capacity of developing E. fluviatilis to readily take up dissolved food sources could be a valuable addition to the daily natural diet of adult stages of the sponge. Evidence is accumulating that many sponge species utilize DOM as major part (71–94%) of their daily carbon intake (Yahel et al., 2003; De Goeij et al., 2008a; Mueller et al., 2014; McMurray et al., 2016), however, these studies were all done on marine sponge species.

The uptake of DOM is hypothesized to be influenced by the presence of associated microbes. It has been shown that mainly sponges with high numbers of associated bacteria (high microbial abundance sponges) utilize DOM as food source, as opposed to the low microbial abundance (LMA) sponges (Hoer et al., 2018), unless sponges have an encrusting growth form (De Goeij, Lesser & Pawlik, 2017). The freshwater sponge E. fluviatilis is a LMA species, but the algal symbionts of E. fluviatilis are not an obligate requirement for survival of the host (Wilkinson, 1980). Moreover, E. fluviatilis can be both encrusting and massive, depending on its stage of development. In this study, we cannot conclude on the role of associated microbes in the processing of DOM. Although all gemmules were sterilized before the experiment, the presence of associated microbes in the sponge specimens cannot be excluded. Moreover, by determining bulk assimilation of the isotope tracer we cannot distinguish whether DOM was processed by symbionts or occurred directly be sponge cells. Future studies using bacterial-and sponge-specific fatty acid biomarkers (De Goeij et al., 2008b, Rix et al., 2017) may identify these processes. It would be interesting to assess whether E. fluviatilis adjusts its diet depending on its algal symbionts and/or at various phases of its life-cycle.

In tropical coral reefs, sponges play a key role in the cycling of nutrients by converting predominantly DOM into particulate organic matter through high cell turnover and detritus production, which is subsequently shunted to higher trophic levels—a process termed the sponge loop (De Goeij et al., 2013). Since detritus production and rapid division of choanocytes by mitosis have been reported for E. fluviatilis (Weissenfels, 1976; Tanaka & Watanabe, 1984), similar sponge-loop processes potentially also occur in freshwater systems. Additionally, the C:N ratio of the amino acids (5:1) in the medium was significantly lower than the assimilated C:N ratio by E. fluviatilis (1:5), and the sponge assimilated up to four times more nitrogen from the amino acids (41%) as compared with amino acid carbon (9%). Carbon could have been partly lost to respiration, but E. fluviatilis most likely assimilated nitrogen selectively from the amino acid source. As sponge detritus in eutrophic tropical ecosystems was found relatively enriched in nitrogen compared to sponge tissue (De Goeij et al., 2013), and the current view is that limitation of nitrogen is also common in eutrophic freshwaters (Downing & McCauley, 1992), freshwater sponges might fertilize their surrounding ecosystems in the same way as their marine counterparts.

Conclusions

Our results demonstrate that the process of skeleton formation in freshwater sponges is highly resilient, coping with extreme g-forces of up to 20 × g for as long as 2 days. These results also support the ideas of Nakayama et al. (2015) that the mechanisms of self-construction shown by E. fluviatilis are adjusted by its micro-environment and are potentially useful in other fields of research such as bioengineering. The underlying cellular and molecular mechanisms of processes such as spicule displacement, arrangement, and cementing as well as their adaptive potential are at present unknown. In addition, this study supports the findings of Skelton & Strand (2013) that freshwater sponges may serve as an important energetic link between pelagic and benthic food webs in systems were sponges are abundant. The uptake of dissolved food may be an important factor in the daily diet of freshwater sponges and they may contribute significantly to the cycling of dissolved nutrients in freshwater ecosystems.

Additional Information and Declarations

Competing Interests

Author Contributions

Data Availability

The authors declare that they have no competing interests.

Martijn C. Bart conceived and designed the experiments, performed the experiments, analyzed the data, prepared figures and/or tables, authored or reviewed drafts of the paper, approved the final draft.

Sebastiaan J. de Vet conceived and designed the experiments, prepared figures and/or tables, authored or reviewed drafts of the paper, approved the final draft.

Didier M. de Bakker performed the experiments, authored or reviewed drafts of the paper, approved the final draft.

Brittany E. Alexander performed the experiments, authored or reviewed drafts of the paper, approved the final draft.

Dick van Oevelen analyzed the data, contributed reagents/materials/analysis tools, authored or reviewed drafts of the paper, approved the final draft.

E. Emiel van Loon analyzed the data, prepared figures and/or tables, approved the final draft.

Jack J.W.A. van Loon conceived and designed the experiments, performed the experiments, contributed reagents/materials/analysis tools, approved the final draft.

Jasper M. de Goeij conceived and designed the experiments, performed the experiments, analyzed the data, contributed reagents/materials/analysis tools, approved the final draft.

The following information was supplied regarding data availability:

https://emielvanloon.github.io/sponge_hypergravity/sponge_hypergravity_report.html

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
