# Peer review of "Spiculous skeleton formation in the freshwater sponge Ephydatia fluviatilis under hypergravity conditions"

_PeerJ, doi:10.7717/peerj.6055_

## Round 0.1 · original submission · Major Revisions

I now have 3 comprehensive reviews from experts in the field of sponge biology. As you will read, all three reviewers asked that a number of issues be addressed in a revision. In particular, reviewer number two provided detailed and substantive issues that need to be addressed. Please revise your manuscript accordingly, and provide a point-by-point response to all of the reviewers comments.

Reviewer 1 ·

Basic reporting

This manuscript has two major components: 1) freshwater sponges are grown from gemmules under hypergravity conditions, 2) freshwater sponges are fed 13C and 15N as a measure of their ability to take up DOM. These two parts of the manuscript are linked by the hypothesis that hypergravity conditions may deplete energy reserves, causing developmental abnormalities, and that nutrient uptake could compensate for this effect.

The manuscript is clearly written, simply designed and appropriately controlled.

Experimental design

The experimental design seems fine. I have only three comments:

1. 13C and 15N enriched medium was added to some sponges, the solution removed and replaced with the regular growth medium. Was there a wash step, or only replacement of the nutrient enriched medium? And if the latter, could the isotopic signatures reflect carry-over of residual nutrient-enriched medium that was not taken up by the sponges? To control for incomplete washing it may have been prudent to add the nutrient-enriched medium to control sponges at the end of the experiment, then immediately remove and replace with regular medium, just to ensure that residual enriched medium did not carry- over as a liquid volume. Also, how were 13C and 15N treatment levels determined? It could be very convincing to titrate this enriched medium to show graduated changes in uptake.

2. Gemmules were pretreated with H202 to minimize fungal and bacterial contamination, but freshwater sponges always have algal and bacterial associates, even after such treatment. How can nutrient uptake be firmly attributed to the sponge and not associated microorganisms?

3. How do g forces relate to ambient pressure. Sponges routinely grow at great depth. Should the expectations for hypergravity could affect sponges be calibrated to account for the fact that sponges have a hydrostatic skeleton? In other words, is 20x g as extreme for a sponge as for an astronaut, or any terrestrial organism? Would it be worth testing a broader range of g force until dramatic effects are detected?

Validity of the findings

No additional comments.

Additional comments

My only real objection to this manuscript is that the two topics don't seem conceptually connected. It seems that the authors have an interest in DOM uptake in sponges, and that they had access to a large diameter centrifuge. Conceptually, the two parts of the study seem largely unrelated and I would probably choose to publish as two discrete manuscripts. Albeit, both may require further development in that case.

Reviewer 2 ·

Basic reporting

1. The entire manuscript should be re-written to fit the general style of a scientific manuscript.

(i) The Introduction section should include sufficient background information to set the work in context. The aims and importance of the work should be clearly stated.

(ii) Results should be presented in a logical sequence in the text, tables and figures. The current description of the results is not detailed enough for readers to understand what is the aim of this experiment, how the experiment was performed and what is each result. Even though the experimental procedures are written in the section of Materials and Methods, the Results section should be written to make it possible for the readers to understand the experiment without reading the Materials and Methods section.

(iii) Figures should be cited in more detail in the text. For example, the current citation is as “(Figure 3, 4)”, but it should be cited as “Figure B upper panel”, or “blue arrows in Figure 3B upper panel”, etc.

(iv) The Discussion section should consider the results in relation to any hypotheses advanced in the Introduction and place the study in the context of other work. To discuss the results in this study, each data should be cited in detail. Currently any data was cited.


2. The English language should be improved and should accurately use correct technical terms to ensure that international readers can clearly understand this manuscript. Some examples are:

(i) hatched gemmules (should be “juvenile sponges hatched from a gemmule”).

(ii) Line 67, degenerative sponge bodies (gemmules), (this is incorrect. Only stem cells are inside the gemmule, and the gemmule is not a degenerative sponge body.

(iii) Line 104 Does “internally-stored energy” mean vitelline platelets that are contained in totipotent stem cells inside the gemmule coat?

(iv) The description “spicules are erected” during the process of skeleton formation is fine. But “sponges erect spicules” or “sponges transport their spicules (line 122)” are scientifically inappropriate descriptions. Since the authors described previous studies of Nakayama et al. as “transport cells” transport spicules and possibly erect spicules, the subjects of these sentences should be “cells” not whole animals (sponges).

(v) So far as I know, “spiculous skeleton” is used instead of “spicule skeleton”.

(vi) Line 129 “relative surface area” needs more explanation; otherwise, it might mislead readers, since the authors seemed to use “surface area” as the dimension of basal epithelia (basopinacoderm) and not as the dimension of the total external surface of the sponge body. Furthermore, what is the meaning of “relative”? what are compared is not clear in this sentence.

3. Previous studies and knowledge should be accurately described. Some examples of the sentences in which this seems to need to be corrected are:

(i) Line 73-77, totipotent stem cells do not differentiate inside the gemmule coat. Stem cells differentiate after they migrate outward from the gemmule coat. What is the difference between “germination” and “gemmule hatching”? During gemmule hatching (development of the juvenile sponge body), stem cells not only attach to the substrate but also differentiate into various types of cells. “cells attach sponge to its substrate” is not an appropriate description. “Cells firstly migrate out from the gemmule coat and differentiate to basal epithelial cells that are attached to the substrate” might be a more accurate description.

(ii) Line 84, So far as I know, it has not been clearly shown that filter systems of juvenile sponges are only developed after spicules are held up. Funayama et al. 2005 showed that choanocyte chambers started to form from developmental stage II. Thus, which phenomenon (spicule holding up or starting the filter system development) is earlier seemed to be difficult to determine at present.

(iii) Line 88, what are the “different locations”? different from what?

(iv) Line 90, what is the “third type of specialized cells?” What are the first and second types of specialized cells? Furthermore, Nakayama et al. clearly reported that basopinacocytes (basal epithelial cells) around the held-up spicules secrete short type collagen to embed the basal end of held-up spicules. Thus, the “third type of specialized cells” in the text should be basopinacocytes around the held-up spicules.

(v) Line 122, Nakayama et al. described that transport cells carry spicule stochastically. Thus, it does not seem likely that transport cells have a certain correct orientation to transport spicules.

4. Only previous studies or knowledge that is directly related to the present study should be described or clearly explained in relation to the present study. An example of a sentence in which the relationship is not clear is:

(i) Line 289, Chitin in the gemmule (coat?) of E. fluviatilis. This does not seem to be related to the skeleton formation.

Experimental design

5. To avoid possibly misleading readers into thinking that Figure 1B shows the results of this study, Figure 1B should be entitled “Our working hypothetical model of the effect of hypergravity (and nutrient uptake) on skeleton construction of juvenile sponge hatched from a gemmule”. Adding a question mark in the illustrated model is commonly used to describe the working hypothesis. Since the current title of Figure 1 is “ Developmental stages of germination in the freshwater sponge, Ephydatia fluviatilis”, Figure 1B could easily be misunderstood as showing the results of this study.

6. The main question in this study should be clearly stated, especially in the Abstract and Introduction. Since the nutrient uptake is described in earlier parts of both sections, it is not clear whether the authors focused on the skeleton construction under hypergravity, or on the effect of nutrient uptake on skeleton construction.

7. Do the data shown in Figure 3 show non-fed juvenile sponges under several different strengths of gravity? If so, it should be clearly described both in the Results section and Figure legend. Photographs of fed-juvenile sponges should be shown, especially since the authors argue that development of the oscula is different between fed- and non-fed juvenile sponges under hypergravity.

8. Does the legend of Table 1 mean that the authors determine the developmental stage of juvenile sponges as “sponges with held-up spicules but without a developed osculum” as Stage III sponges and “sponges with a developed osculum” as stage IV sponges?

Validity of the findings

9. In the Discussion section, several arguments or conclusions of authors seemed to be overstatements, or I could not follow the authors’ scientific logic. This might be partially because of the inappropriate wording. For example:

(i) Line 343 (the first sentence of the conclusion). Since this study did not examine any of “the process of skeleton formation”, current statement is overstatement. Not the process but whether the skeleton is formed (spicules are held up) is examined in this study.

(ii) Line 277-278, why does the result that juvenile sponge could be developed under the hypergravity forces show that a pivotal step in the seasonal life-cycle of E. fluviatilis is highly “resilient”? If “a pivotal step in the seasonal life-cycle” means development of a juvenile sponge from a gemmule, I would like to suggest that the authors cite the literature that describes that gemmule hatching is seasonal in this species. So far as I know, Ephydatia fluviatilis generally can survive winter as adult sponges, and gemmule hatching is not really included their life-cycle. In contrast, the sister species Ephydatia muelleri generally survives winter as gemmules (and thus gemmule hatching is included in its seasonal life-cycle).

(iii) Line 282. the authors described that sponges showed a significant “decrease” in surface area under increased g-force exposure But in Figure 4, all specimens showed increase (>100%). It is confusing and could not follow what the authors would like to describe.

(iv) Line 303- Since only juvenile sponges in the process of developing the body, including the filter-feeding system, were used in this study, it is difficult to distinguish whether the results suggest that the process of skeleton construction relies mainly on intrinsic energy or whether they suggest that juvenile sponges simply could not take up sufficient nutrients to affect skeleton construction in these developmental stages.

(v) Line 306- it is interesting that development of an oscule was significantly less frequent in fed juvenile sponges, especially under hypergravity force. but the data did not show or describe this effect (numbers, etc.). So, it is difficult for readers to evaluate how “significant” the effect was.

Additional comments

1. One of the major problems is that the authors did not show data concerning whether there is any difference of body growth between non-fed and fed juvenile sponges in the general culture condition (gravity 1g). I suppose that juvenile sponges grow similarly in non-fed or fed conditions, since in the working hypothesis in Figure 1B, in 2.5g the illustrations of the morphology of juvenile sponges show that they are the same.
If so, this suggests that in juvenile sponges from developmental stage II to IV (stage IV is the stage when the oscule forms and the filter-feeding system becomes fully functional) there is not much uptake of nutrients from the culture medium. The fact that the filter-feeding system is not completed (without an osculum to expel water) in the developmental stages the authors used in this study also supports this possibility. Thus, it is difficult to follow the logic of the authors’ working hypothesis to expect difference between non-fed and fed conditions.
Since the experiments themselves using hyper gravity are interesting, I suggest that the authors re-consider the stream and logic of the manuscript.

2. Although the title of this manuscript is “Spicule skeleton formation in the freshwater sponge Ephydatia fluviatilis under hypergravity conditions”, it was not clear the main question of this study is whether the authors focused on the skeleton construction under hypergravity, or on the effect of nutrient uptake on skeleton construction. As there are quite a lot of description about nutrient uptake in Abstract, Introduction and Discussion, it is very confusing and difficult to follow the main stream of this manuscript.

3. The entire manuscript should be re-written more accurately. There are many inappropriate descriptions of previous knowledge and works in Introduction section. Furthermore, many parts of Discussion are overstated, especially conclusion. Conclusion should be based on the results of this work.

Reviewer 3 ·

Basic reporting

The English writing could be improved

Experimental design

OK - no comment

Validity of the findings

no comment

Additional comments

PeerJ #30193

Evaluation of the manuscript “Spicule skeleton formation in the freshwater sponge Ephydatia fluviatilis under hypergravity conditions” by Bart et al.

This manuscript describes a work, which fits publication in PeerJ. The authors examined growth and formation of a 3D skeleton in freshwater sponge under hypergravity conditions. They assumed that under such conditions is energetically more costly than under normal g force, and hence with increasing g-force the growth will be slowed down. They examined this assumption using growth post germination of freshwater sponge gemmules. Furthermore, in order to evaluate if lack of energy is driving such hypothesized phenomenon, they supplied half of the gemmules with DOM (dissolved organic matter) in the form of (C and N) marked amino acids. I liked the experiment using the Large Diameter Centrifuge designed to examine behavior/physiology under various gravity forces.

Some minor comments are given below – but I consider the manuscript suitable for publication.

There are comments I do not write which I assume are the result of converting the text to a pdf format (many places were space is required between words).
The number of specimens treated (n) should be given in all places relevant (including figures 4 & 5).

The other remarks are written in order of appearance in the text.

L. 73 - Ilan et al., 1996 does not appear in the references
L. 93 - Nakayama (2015), should be Nakayama et al. (2015),
L. 277 - E. Fluviatilis should be E. fluviatilis
L. 278 - ” under extreme levels of mechanical stress” What do you consider to be “extreme levels”?
L. 285 - sponging should be spongin
L. 285 - Regarding chitin in adult E. fluviatilis - I am not aware of chitin reported from mature Ephydatia. See: PloS one 13.5 (2018): e0195803.
L. 287 - “ invertebrates and marine demosponges” implies that sponges are not invertebrates. I would change to “invertebrates including marine demosponges”
L. 298-301 – What about adult E. fluviatilis microbiota?
L. 308 - prologue should be prolonged.
L. 316 - McMurray 2016 should be McMurray et al., 2016
L. 355 - DOM uptake was shown here in a developing sponge - not in an adult.

---

## Round 0.2 · accepted · Accept

I believe the authors have satisfactorily addressed the comments of the reviewers and that the revised manuscript is ready for publication.

#